# Cluster Analysis of Factors Associated with Leishmaniasis in Peru

**DOI:** 10.3390/tropicalmed8110484

**Published:** 2023-10-26

**Authors:** Irma Luz Yupari-Azabache, Jorge Luis Díaz-Ortega, Lucía Beatriz Bardales-Aguirre, Shamir Barros-Sevillano, Susana Edita Paredes-Díaz

**Affiliations:** 1Grupo de Investigación en Enfermedades Infecciosas y Transmisibles, Universidad César Vallejo, Av. Larco 1770, Trujillo 13009, Peru; jdiazo@ucv.edu.pe (J.L.D.-O.); shamir.bs17@gmail.com (S.B.-S.); sparedes@ucv.edu.pe (S.E.P.-D.); 2Escuela Profesional de Nutrición, Universidad César Vallejo, Trujillo 13001, Peru; 3Departamento de Ciencias, Universidad Privada del Norte, Trujillo 13001, Peru; lucia.bardales@upn.edu.pe; 4Sociedad Científica de Estudiantes de Medicina de la Universidad César Vallejo, Trujillo 13001, Peru

**Keywords:** leishmaniasis, factors, cluster analysis

## Abstract

Objective: To analyze the factors associated with leishmaniasis in Peru, according to the cluster classification in the period 2017–2021. Methods: Quantitative approach, with an ecological, descriptive correlational, and cross-sectional design. The population was from the geographical region of Peru, where a total of 26,956 cases of leishmaniasis were registered by the Peruvian Ministry of Health from 2017 to 2021. Spearman’s Rho statistic was used to analyze the variables that are most associated with the cases of leishmaniasis reported per year, and, in addition, the multivariate technique of cluster analysis was applied. Results: Annual rainfall and areas with humid forest (climatic factors) and mortality from transmissible diseases (health factor) are directly associated with reported cases of leishmaniasis. Households with basic access to infrastructure, drinking water, drainage, and electric lighting; illiteracy, regional social progress, and unsatisfied basic needs (social factors); and percentage of urban population (demographic factor) are inversely and significantly associated with cases of leishmaniasis. Conclusions: Climatic and environmental factors contribute to the multiplication of the leishmaniasis disease vector. The incidence of leishmaniasis adds up to the mortality rates for transmissible diseases in Peru. As living conditions improve, the incidence of this pathology decreases.

## 1. Introduction

Leishmaniasis is a parasitic disease of worldwide distribution [1], caused by a protozoan parasite that belongs to the family Trypanosomatidae called Leishmania [2]. Leishmaniasis is endemic in 98 tropical and subtropical countries, putting more than 350 million people at risk, being projected to have almost two million new cases per year [3]. Peru is among the 10 countries in the world with the highest number of cases of cutaneous leishmaniasis and among the four highest in Latin America, along with Brazil, Colombia, and Nicaragua [4].

Leishmaniasis is a chronic disease caused by flagellated protozoa of the genus Leishmania, of which there are more than 20 species and about 90 vectors involved in the transmission of the parasite [5].

Leishmania parasites are obligate intracellular, transmitted by the bite of infected female sandflies of the genera *Phlebotomus* and *Lutzomyia*. Leishmaniasis is essentially a zoonotic disease, in which the reservoir varies according to the Leishmania species. *Leishmania donovani* have humans as one of the reservoirs, while dogs are the reservoir of the species *Infantum* and *Infantum chagasi*. Likewise, rodents are the main reservoir of the species *Major*, *Mexicana*, *Amazonensis*, and *Braziliensis*, these occurring mainly in South America [6,7].

The latency period of leishmaniasis depends on the species of leishmania or the patient’s immune system and can vary from weeks to months [6]. Cutaneous leishmaniasis is the most frequent, producing ulcerative lesions with lifelong scars in most cases [8]. This pathology varies depending on the region in which it occurs. In visceral leishmaniasis, hepatosplenomegaly, lymphadenopathy, and weight loss are established. It is mainly caused by *L. donovani* in adults in eastern Africa and Asia, especially in India, with skin sequelae known as post-kala-azar, but it is also present in children via *L. infantum chagasi* [6]. Among the main risk factors for visceral leishmaniasis are malnutrition and immunosuppression caused by human immunodeficiency virus (HIV) [4]. About 95% of cases of cutaneous leishmaniasis occur in South America, the Mediterranean Basin, the Middle East, and Central Asia. Most cases of visceral leishmaniasis occur in Brazil, East Africa, and India; likewise, more than 90% of the cases of mucocutaneous leishmaniasis occur in Brazil, Bolivia, Ethiopia, and Peru [4].

Leishmaniasis is neglected worldwide, thus becoming a serious public health problem, due to its very disabling and disfiguring consequences, seriously impacting the economic wellbeing and mental health of those people exposed to it and health services, through requiring high-cost treatment [9].

The risk factors in the transmission of leishmaniasis are multiple, and socioeconomic conditions such as poverty, poor housing conditions, and poor sanitation promote the faster transmission of this parasite. Likewise, it is mentioned that the incidence of leishmaniasis can increase due to the presence of malnutrition, population mobility, and environmental and global climatic change. Knowing these factors makes it possible to prevent and develop strategies that contribute to its reduction, since transmission takes place in a “complex biological system that includes the human host or animal reservoir, the parasite, and the sandfly vector” [4].

In Peru, cutaneous and mucocutaneous leishmaniasis are present; there is no visceral leishmaniasis. Amazonian mucocutaneous leishmaniasis or “espundia” occurs in areas with altitudes below 1800 m, while the Andean cutaneous or “uta” is endemic in western and inter-Andean valleys of northern and central Peru between 1000 and 3200 m altitude [10].

In Peru, only a small number of Lutzomyia (Lu.) species have been identified as vectors of cutaneous leishmaniasis, including *Lu. peruensis*, *Lu. verrucarum*, *Lu. tejada*, *Lu. Ayacuchensis*, and *Lu. pescei*. In the Amazon region, Lutzomyia species of the subgenus *Viannia* have been found naturally infected with Leishmania, *yuilli yuilli*, *Lu. Chagasi*, *Lu. Davisi*, and *Lu. auraensis* [11].

In the Andean region of Peru, cutaneous leishmaniasis caused by *Leishmania (Viannia) peruviana* is predominant, while in Amazonia, mucocutaneous leishmaniasis is mainly caused by *Leishmania (Viannia) braziliensis*, in which the parasite metastasizes from an initial skin lesion towards the naso-oropharyngeal mucosa and affects 20–40% of patients; however, the constant incidence in recent years of this clinical manifestation is due to the presence of a hybrid species of *L. brazilensis*/*L. peruviana*, whose identified vector was Lutzomyia *tejadai* in the city of Huánuco [11].

In 2022, the Peruvian Ministry of Health reported that, from week 01 to week 24, 2343 cases of leishmaniasis had been reported in the country, without deaths, while in the same period in 2021, 3187 cases were reported [12].

The presence of a vector is not the only factor that determines whether or not a pathogen can be established. In particular, local environmental components (i.e., climate, host foraging, presence of vertebrate reservoirs, and accessibility to humans) represent factors influencing pathogen amplification and spread of pathogen-borne diseases [13].

Temperature and humidity have effects on the development of sandflies as vectors of cutaneous leishmaniasis [14]. Higher temperatures favor the multiplication of these insects while increasing their activity and frequency, which reinforces the vectorial capacity of sandflies and the transmission of leishmaniasis. However, precipitation inhibits their proliferation [15]. It has been found that the ranges of temperature and relative humidity for the development of sandflies are between 21–29 °C and 80–90%, respectively [16]. Also, within environmental factors, it has been reported that vegetation and altitude play especially important roles in determining the habitats and vectorial capacities of sandflies [17]. 

Socioeconomic factors and ecosystem changes have been linked to impacts on both cutaneous and visceral leishmaniasis, including urbanization, deforestation, agricultural intensification (dams and irrigation, new crops), human settlements (including rural-to-urban migration), poverty and marginalization, and the development of new crops [18]. 

Among protective factors against the disease, the presence of basic services and the knowledge of the disease on the part of those individuals studied are considered. It has been demonstrated that having adequate knowledge of the disease is fundamental to reducing the risk of acquiring it, in the case of cutaneous leishmaniasis [5].

In this sense, due to the consequences and because these basically affect the most vulnerable populations, it is considered necessary to know these risk factors and classify them within a certain population.

Several studies have been conducted using cluster analysis. A previous study in Spain showed that the cases of leishmaniasis appeared to be distributed around the main urban centers [19], leading to proposed hypotheses about probable etiologies and recommendations for more focused surveillance in the city.

On the other hand, from 4951 municipalities spread throughout Latin America, it was possible to define seven clusters based on their correlation with 18 socioeconomic and environmental characteristics. They also found that the Amazonian, Andean, and Sabanero ethnic groups in Latin America had positively and consistently increased historical risks of cutaneous leishmaniasis [20]. This helped to understand the epidemiological pattern of transmission on a larger scale, and it suggests actions for the health sector, as well as for many other sectors, for the prevention of the disease in the Latin American region. But despite having a high prevalence of the disease and having the potential to obtain great benefits from research, there are no studies of this type in Peru; for this reason, the present investigation aims to analyze the climatic, environmental, social, demographic, and health factors and conditions associated with leishmaniasis in Peru, according to the cluster classification in the period 2017–2021.

## 2. Materials and Methods

The present study is a quantitative approach, with an ecological, correlational, and cross-sectional design [21].

### 2.1. Population, Sample, and Sampling

The population was constituted according to the 25 regions of Peru, including the Callao region, where a total of 26,956 cases of leishmaniasis were reported by the Peruvian Ministry of Health (MINSA) during the years 2017 to 2021. No sampling techniques were applied, since all the data found were used. MINSA, through its transparency portal (accessed on 6 March 2023, https://cutt.ly/Rww7dOHF), provided complete information on all the patients, without making any type of exception. Therefore, the results helped to understand the problem at a national level [22].

For the data collection technique, documentary analysis of databases and/or epidemiological records was used, and the instrument used was a simple data collection form prepared by the authors. This data collection form is divided into four sections.

The first section identifies the climatic and environmental factors of the reported cases of leishmaniasis for each region. The variables included were average annual maximum temperature, average annual temperature, average annual relative humidity, total annual precipitation, and percentage of the region with Amazonian rainforest.

The second section identifies the social factors of the reported cases of leishmaniasis. The variables included were the percentage of households with basic access to infrastructure, drinking water, access to sewage and sanitation, access to electric lighting, districts with pollutants, illiteracy rates of the population aged 15 years and older, percentages of poverty, regional social progress index, overcrowding, unsatisfied basic needs, and the human development indicator.

The third section identifies the demographic factors of the reported cases of leishmaniasis, including variables such as life expectancy, population density, age dependency ratio, population over 64 years of age, and percentage of urban population.

The fourth section identifies health factors of the reported cases of leishmaniasis, including variables such as mortality due to transmissible diseases per 1000 inhabitants and number of physicians per 1000 inhabitants.

All these data were extracted from the web portal of the National Institute of Statistics and Informatics (accessed on 8 March 2023, www.inei.gob.pe).

### 2.2. Statistical Analysis

For the processing of the information, the free-use statistical program JAMOVI version 2.3.21was used [23,24]. Likewise, to identify the climatic, environmental, social, demographic, and health factors by region, an exploratory analysis of the data was applied in the first instance using descriptive statistical measures and in the inferential statistical analysis through the normality tests of the data to test the normality assumption.

As part of the exploratory data analysis, the assumption of normality was tested, and, since it was found that the data are not normally distributed, the Spearman’s Rho statistic was used to analyze the variables that are most associated with reported cases of leishmaniasis. Subsequently, the variables that showed a moderate or strong statistical association and were found to be significant were considered for continued application to the multivariate cluster analysis technique to identify the factors that best explain the behavior of leishmaniasis and detect groups that formed with greater similarities. In addition, statistical measures such as mean as a measure of central tendency and standard deviation as a measure of dispersion were presented. The exploratory data analysis was complemented with box and whisker plots carried out with IBM SPSS software version 25 classified by cluster for each considered study variable [22,25].

Cluster analysis is a multivariate exploratory analysis method that allows for the classification of study units, without requiring the verification of prior assumptions such as normality and stationarity, among others. It allows for the formation of clusters according to similarities within the same group and significant differences between them [26]. Taking into account the characteristics of cluster analysis, this method was used to classify the regions of Peru, considering the variables of the present study. Standardized data were processed, and Euclidean distance measures and Ward’s clustering method were applied as hierarchical grouping methods. In addition, dendrogram and phylogenetic graphs were used for better visualization of the formed clusters [26].

Finally, the clusters formed by the regions of Peru were represented on a cartogram, created using the National Platform of Georeferenced Data “Geo Peru” (accessed on 1 July 2023, https://visor.geoperu.gob.pe/).

### 2.3. Ethical Considerations

This research was based on the ethical principles of non-maleficence, autonomy, beneficence, and respect. Confidentiality and identity were respected, as the data were organized in codes. It was not necessary to request permission or informed consent, since the data are freely published on the web page of the Peruvian Ministry of Health. Likewise, this work was submitted for evaluation to the Ethics Committee of the School of Medicine of the Universidad César Vallejo, which approved it by means of resolution 015-CEI-EPM-UCV-2023.

## 3. Results

The results of 26,956 cases of leishmaniasis that occurred in the regions of Peru and were reported by the Ministry of Health during the years 2017 to 2021 were the processed regions.

Table 1 shows that, of the climatic and environmental factors analyzed, those associated with the reported cases of leishmaniasis in Peru were the total annual precipitation (mm) and the percentage of the region with Amazonian humid forest, both of which were associated in a moderate and direct way in the periods from 2017 to 2021, being significant except for the year 2019 in the case of total annual rainfall. On the other hand, regarding social factors, it is observed that the percentage of households with basic access to infrastructure, percentage of households with drinking water, percentage of households with access to drainage–sanitation, percentage of households with access to electric lighting, and the Regional Social Progress Index is inversely and moderately associated with the reported cases of leishmaniasis, allowing us to deduce that as we find better living conditions, there are fewer reported cases of leishmaniasis in Peru during the study periods. On the contrary, the illiteracy rate of the population aged 15 and over and the percentage of households with unsatisfied basic needs were directly and moderately associated, demonstrating that, at higher illiteracy rates, the reported cases of leishmaniasis are higher during the period 2017–2021, with the results being significant (*p* < 0.05).

Likewise, when analyzing demographic factors, it can be seen that life expectancy (in years), the percentage of the population older than 64 years, and the percentage of urban population were inversely associated; however, only age greater than 64 years and the condition of having an urban population were considered with significant associations in almost the entire study period; this indicates that the regions of Peru that have longer-living populations and are located in urban areas will have lower numbers of cases of leishmaniasis infection.

Finally, Table 1 shows that the cases of mortality from communicable diseases per 100,000 inhabitants are directly associated with the reported cases of leishmaniasis, which enables the deduction that this communicable disease contributes to raising the indicators of mortality from communicable diseases in Peru; it should be noted that this association was considered direct and moderate, in addition to being significant during almost the entire study period, with the exception of 2019.

Figure 1 shows that when applying the cluster analysis with Ward’s method and the Euclidean distance measure to the data from the regions of Peru, considering the indicators that were significantly associated with the reported cases of leishmaniasis to the year 2021, four well-differentiated clusters were obtained, which were composed as follows: Cluster 1 (C1) with nine regions: Ancash, Arequipa, Callao, Ica, La Libertad, Lambayeque, Lima, Moquegua, and Tacna; Cluster 2 (C2) with three regions: Loreto, Madre de Dios, and Ucayali; Cluster 3 (C3) made up of four regions: Amazonas, Huánuco, Pasco, and San Martín; and Cluster 4 (C4) made up of nine regions: Apurímac, Ayacucho, Cajamarca, Cusco, Huancavelica, Junín, Piura, Puno, and Tumbes. These clusters are differentiated by colors in figures (A) as a dendrogram and (B) as a phylogenetic network of the regions of Peru.

From Table 2 and Figure 2, it can be said that C1 is made up of the regions with the lowest number of reported cases of leishmaniasis, in addition to presenting the lowest indexes of total annual rainfall, quite close to those presented by C4. In addition, the cluster of regions with the highest average number of leishmaniasis cases reported was C2, the same cluster with the highest amount of rainfall, showing that high rainfall values can generate an increase in the transmission vector of leishmaniasis, and therefore there is an increase in the number of cases reported, similar to those presented by C3. Likewise, in figure (A) we can see the box diagram for confirmed cases of leishmaniasis and in figure (B) the total annual rainfall, noting what was previously described.

Figure 3 is divided into figure (A), where it shows a box diagram of the leishmaniasis cases, and from figures (B) to (H), the indicators of the social factors. Analyzing these, as well as Table 2, it can be said that C1 is made up of the regions with the least number of reported leishmaniasis cases, in addition to presenting the best figures for basic access to infrastructure, drinking water, drainage and sanitation, and lighting and Electricity, as well as the regional social progress index. C4 is quite heterogeneous in terms of the number of confirmed cases of leishmaniasis; however, it is fairly stable on the basic access flags mentioned above. Clusters 2 and 3 are those that present indicators of precarious access to basic services at home, going hand in hand with the high indicators of confirmed cases of leishmaniasis. The educational component measured by the illiteracy rate shows that the clusters with lower illiteracy rates have fewer reported cases of leishmaniasis.

In Table 2, it can also be seen that, in the demographic aspect, C1 is made up of the regions with the least number of reported leishmaniasis cases, also presenting higher values of the dependent adult population and greater urban population; C2, however, shows less population over 64 years of age regardless of the area where they reside (urban or rural) and an increase in cases of leishmaniasis.

In Figure 4, divided into figures A, B, and C, we can see that Cluster 1 is characterized by presenting elderly populations mainly located in urban areas, but with low numbers of confirmed cases of leishmaniasis. Cluster 2 has a higher number of confirmed cases of leishmaniasis; however, it has a smaller population of people over 64 years of age, and more homogeneous data (less dispersed), and people are located predominantly in urban areas, but to a lesser extent than Cluster 2. Cluster 3 shows a low dispersion of confirmed cases of leishmaniasis, bordering on 207 cases. In addition, the concentration of the population of adults over 64 years of age is around 6.1 cases on average, this group of regions being the most homogeneous. Likewise, it has a greater dispersion of regions located in urban areas than the other clusters. Finally, with Cluster 4, although it is true that it has an average of 230 cases, this cluster is quite dispersed, however, in the number of reported cases of leishmaniasis compared to the other clusters.

Figure 5 was divided into two, where figure A shows the number of confirmed cases of leishmaniasis, and figure B shows the mortality rate from communicable diseases. From these figures, as well as from analyzing the health factors in Table 2, we can see that the mortality rates in each cluster are directly proportional to the reported leishmaniasis cases.

Finally, in Figure 6, it can be seen that when representing the clusters in a cartogram, this clearly denotes that the regions that make up the most affected clusters (C2 and C3) are those in which there is less access and state intervention, in addition to having tropical climates and being located in the Amazonian region of Peru. The regions that are in regular conditions are those that make up the northeast region and those whose access to services is limited. Finally, the regions of the north, center, and south coast of the country (C1 and C4) have good indicators, showing favorable climate figures, good social indicators of access to services and education, and satisfactory demographic and health indicators.

## 4. Discussion

The results indicate that the total annual precipitation (mm) and the percentage of the area with Amazonian humid forest were associated with cases of leishmaniasis, confirming that the impact of climatic factors on ecological conditions contributes to the multiplication of the leishmaniasis vector disease [15,16,17]. Similarly, in a study carried out in Argentina, the results presented on the eco-epidemiology of leishmaniasis showed that climatic changes result in changes in the probability of effective vector–human reservoir contacts [27].

Likewise, this is observed in other countries with humid forest areas such as Manabi in Ecuador, where outbreaks have been associated with human incursion into a wild habitat with natural transmission such as clearing forests, road construction, or new colonization areas, or in the case of migration to endemic areas to carry out work such as gold or wood extraction or coffee harvesting [28].

The results during the study periods also indicate that with better living conditions, there are fewer reported cases of leishmaniasis in Peru, and that, at higher illiteracy rates, the reported cases of leishmaniasis are higher. This confirms the results reported in a study carried out in South America and Mesoamerica, where socioeconomic factors such as inadequate living conditions in human settlements have an impact on the presence of the leishmaniasis vector [18].

Age is also a factor associated with the presence of leishmaniasis, with most cases relapsing in people over 64 years of age; however, long-living people located in urban areas have a lower chance of contracting this disease. This differs from a study carried out in other areas, such as Saudi Arabia, where the highest concentration of patients was grouped in young ages, especially children from 0 to 10 years [29]. Likewise, some researchers indicate that leishmaniasis can occur in any age range; some national and international studies show a higher incidence in the young or young adult groups [29,30].

The division of the regions into four clusters highlights the presence of factors associated with this disease, so the C1 regions of Lambayeque, La Libertad, Ancash, Lima (capital of Peru), Ica, Arequipa, Moquegua, and Tacna have better living conditions and a better rate of social progress; likewise, there are fewer confirmed cases according to MINSA statistics [12].

However, in Clusters 2 and 3 there are indicators of precarious access to basic services at home, as well as climatic conditions favorable to the development of this vector, with a greater number of confirmed cases of leishmaniasis. The regions of Amazonas, San Martin, Huánuco, Pasco, Loreto, Ucayali, and Madre de Dios are located here, constituting the Amazonian area of Peru.

These clusters help to differentiate the cities where there is a higher probability of occurrence of leishmaniasis and to know the factors that contribute to it. Similarly, another study conducted in Latin America observed the presence of seven clusters for the characterization of cutaneous leishmaniasis, taking into account socioeconomic variables related to sanitation and education, as well as basic household services such as sanitation and water, keys for the establishment of said problem. Similarly, coinciding with the results here, it took into account environmental factors such as temperature and precipitation [20].

On the other hand, a study carried out in Peru reflects the results here, since it indicates that there are areas where unusual increases in the number of cases have been observed since the 1980s and 1990s, for example, in the region of Huánuco, where the presence of the inter-Andean Huallaga valley in Ambo province has an open-valley ecology connected to Amazonia, allowing for the presence of both Andean and Amazonian leishmaniasis [11].

Regarding the limitations of the study, it can be mentioned that there was access to the complete data of the epidemiological files, for which the information provided by MINSA through the transparency portal was adapted, saving the necessary variables for the investigation. In the social aspect, there was no information on inhabitants who were engaged in activities related to agriculture, nor on their main occupation; there was also no information on the presence of animals in the home; these aspects have also been associated in other studies [31,32,33].

However, the classification and identification of the factors associated with the presence of leishmaniasis in this study contribute to increasing knowledge about the behavior of this vector in such a way that it supports authorities when making decisions for the prevention of the spread of this disease.

## 5. Conclusions

Climatic, environmental, social, demographic, and health factors were moderately and directly associated with reported cases of leishmaniasis in Peru during the period from 2017 to 2021. On the other hand, social factors were inversely and moderately associated. Cluster 2 (regions of Loreto, Madre de Dios, and Ucayali) and Cluster 3 (regions of Amazonas, Huánuco, Pasco, and San Martín) are those most affected by leishmaniasis, both due to climatic conditions and the lack of government intervention Finally, the incidence of leishmaniasis adds to the mortality rates for transmissible diseases in Peru, and as living conditions improve, the incidence of this pathology decreases.

## Figures and Tables

**Figure 1 tropicalmed-08-00484-f001:**
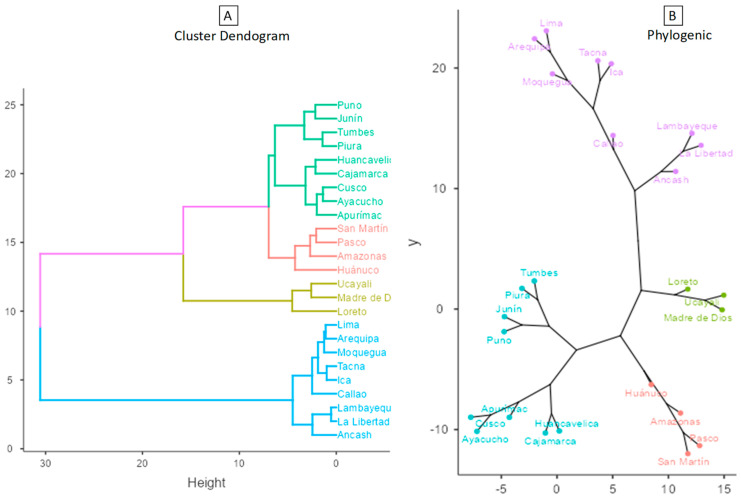
Dendrogram and Phylogenic network of the regions of Peru (Ward’s method–Euclidean distance).

**Figure 2 tropicalmed-08-00484-f002:**
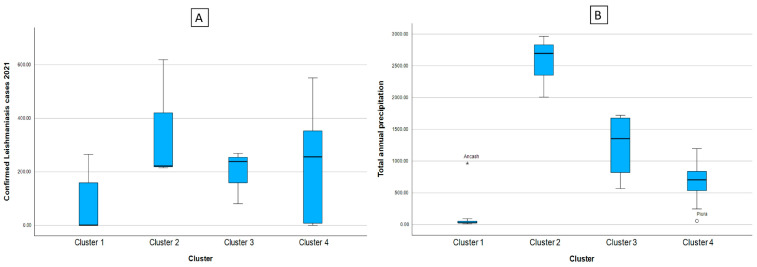
Box diagram of the cases of leishmaniasis (**A**) and indicators of the climate and environmental factors according to cluster (**B**). Note: * indicates the existence of atypical values.

**Figure 3 tropicalmed-08-00484-f003:**
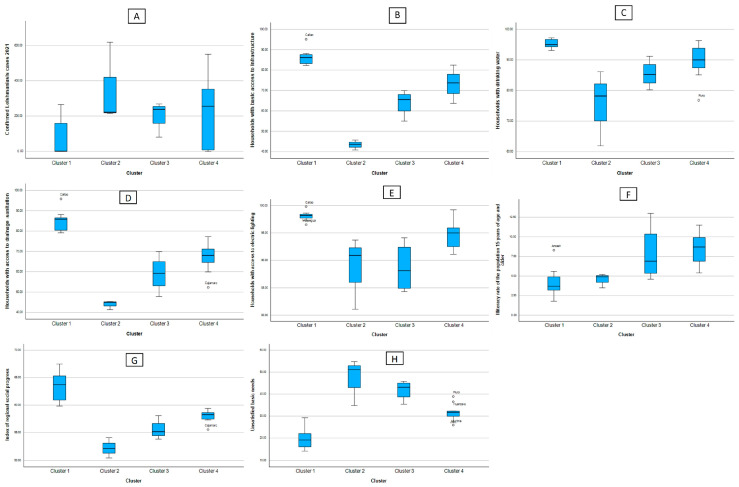
Box diagram of the cases of leishmaniasis (**A**) and indicators of the social factors according to cluster. (**B**) Households with basic access to infrastructure (**C**) Households with drinking water (**D**) Households with access to drainagesanitation (**E**) Households with access to electric lighting (**F**) Illiteracy rate of the population 15 years of age and older (**G**) Index of regional social progress (**H**) Unsatisfied basic needs.

**Figure 4 tropicalmed-08-00484-f004:**
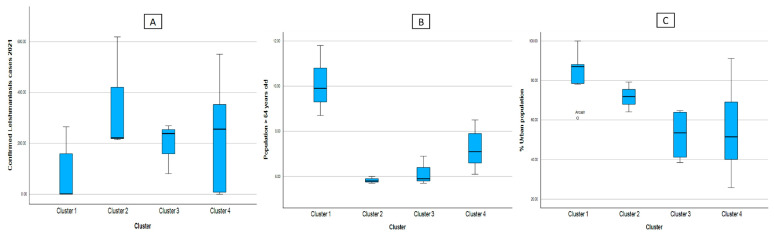
Box diagram of the cases of leishmaniasis (**A**) and indicators of the demographic factors according to cluster, (**B**) Population > 64 years old, (**C**) %Urban population.

**Figure 5 tropicalmed-08-00484-f005:**
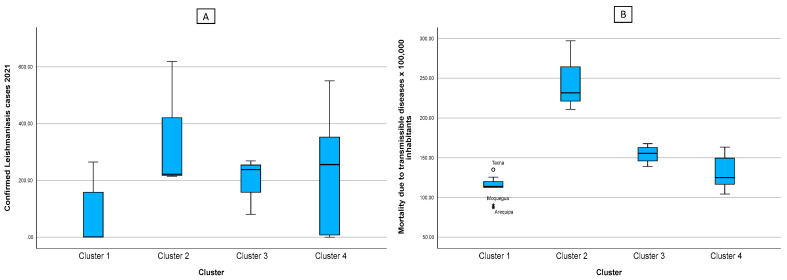
Box diagram of the cases of leishmaniasis (**A**) and indicators of the health factor according to cluster (**B**). Note: * indicates the existence of atypical values.

**Figure 6 tropicalmed-08-00484-f006:**
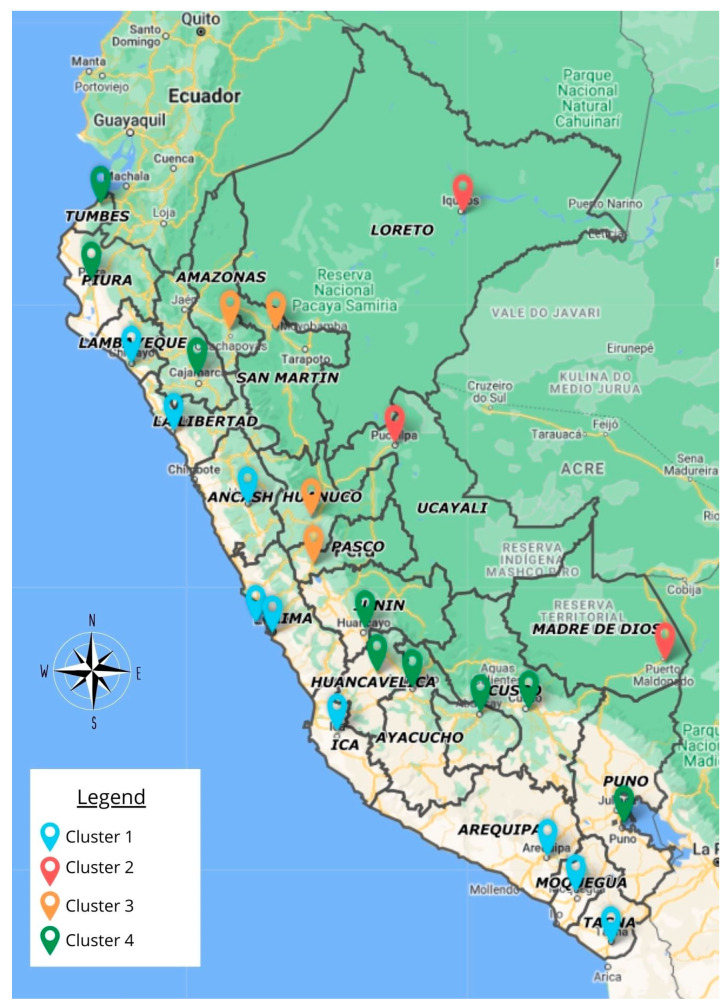
Geographical distribution of the profile of leishmaniasis cases and their associated factors, according to cluster, in 2021.

**Table 1 tropicalmed-08-00484-t001:** Factors associated with confirmed cases of leishmaniasis in Peru, in the period 2017–2021.

Factor	Indicator		2017	2018	2019	2020	2021
Climatic and environmental	Maximum annual average temperature	R	−0.09^d^	−0.08 ^d^	−0.005 ^d^	−0.075 ^d^	−0.075 ^d^
		*p* value	0.668	0.703	0.982	0.723	0.723
	Annual average temperature	R	−0.058 ^d^	−0.037 ^d^	0.012 ^d^	−0.039 ^d^	−0.039 ^d^
		*p* value	0.785	0.863	0.955	0.855	0.855
	Annual average relative humidity	R	−0.026 ^d^	−0.007 ^d^	−0.029 ^d^	−0.093 ^d^	−0.093 ^d^
		*p* value	0.902	0.973	0.889	0.659	0.659
	Total annual precipitation	R	0.51 ^b^	0.436 ^b^	0.344 ^c^	0.399 ^c^	0.399 ^c^
		*p* value	0.009 *	0.030 *	0.092	0.048 *	0.048 *
	% of the region with Amazon rainforest	R	0.746^a^	0.72^a^	0.567^b^	0.717^a^	0.717^a^
		*p* value	<0.0001 *	<0.0001 *	0.003 *	<0.0001 *	<0.0001 *
Social	Households with basic access to infrastructure	R	−0.646 ^a^	−0.628 ^a^	−0.534^b^	−0.638^a^	−0.638^a^
		*p* value	0.001 *	0.001 *	0.006 *	0.001 *	0.001 *
	Households with drinking water	R	−0.587 ^b^	−0.556 ^b^	−0.47 ^b^	−0.583 ^b^	−0.583 ^b^
		*p* value	0.002 *	0.004 *	0.018 *	0.002 *	0.002 *
	Households with access to drainage–sanitation	R	−0.663 ^a^	−0.649 ^a^	−0.557 ^b^	−0.647 ^a^	−0.647 ^a^
		*p* value	0.000 *	0.000 *	0.004 *	0.001 *	0.001 *
	Households with access to electric lighting	R	−0.501 ^b^	−0.457 ^b^	−0.324 ^c^	−0.48 ^b^	−0.48 ^b^
		*p* value	0.011 *	0.022 *	0.114	0.015 *	0.015 *
	Districts with polluting elements	R	0.32 ^c^	0.312 ^c^	0.432 ^b^	0.37 ^c^	0.37 ^c^
		*p* value	0.119	0.129	0.031 *	0.069	0.069
	Illiteracy rate of the population 15 years of age and older	R	0.446 ^b^	0.444 ^b^	0.461 ^b^	0.457 ^b^	0.457 ^b^
		*p* value	0.026 *	0.026 *	0.020 *	0.022 *	0.022 *
	% Poverty	R	0.164 ^d^	0.204 ^c^	0.135 ^d^	0.222 ^c^	0.222 ^c^
		*p* value	0.435	0.328	0.521	0.287	0.287
	Index of regional social progress	R	−0.693 ^a^	−0.663 ^a^	−0.542 ^b^	−0.658 ^a^	−0.658 ^a^
		*p* value	0.000 *	0.000 *	0.005 *	0.000 *	0.000 *
	Overcrowding	R	0.423 ^b^	0.38 ^c^	0.241 ^d^	0.319 ^c^	0.319 ^c^
		*p* value	0.035 *	0.061	0.246	0.120	0.120
	Unsatisfied basic needs	R	0.491 ^b^	0.449 ^b^	0.349 ^c^	0.434 ^b^	0.434 ^b^
		*p* value	0.013 *	0.024 *	0.087	0.030 *	0.030 *
	Human development indicator	R	−0.354 ^c^	−0.34 ^c^	−0.198 ^d^	−0.325 ^c^	−0.325 ^c^
		*p* value	0.083	0.096	0.343	0.113	0.113
Demographic	Life expectancy	R	−0.431 ^b^	−0.389 ^c^	−0.214 ^c^	−0.336 ^c^	−0.336 ^c^
		*p* value	0.031 *	0.054	0.305	0.101	0.101
	Population density	R	−0.262 ^c^	−0.233 ^c^	−0.079 ^d^	−0.216 ^c^	−0.216 ^c^
		*p* value	0.206	0.263	0.707	0.299	0.299
	Dependency ratio by age	R	0.219 ^c^	0.205 ^c^	0.093 ^d^	0.186 ^d^	0.186 ^d^
		*p* value	0.294	0.326	0.660	0.374	0.374
	Population > 64 years old	R	−0.534 ^b^	−0.51 ^b^	−0.378 ^c^	−0.493 ^b^	−0.493 ^b^
		*p* value	0.006 *	0.009 *	0.062	0.012 *	0.012 *
	% Urban population	R	−0.523 ^b^	−0.506 ^b^	−0.44 ^b^	−0.507 ^b^	−0.507 ^b^
		*p* value	0.007 *	0.010 *	0.028 *	0.010 *	0.010 *
Health	Mortality due to transmissible diseases per 100,000 inhabitants	R	0.465 ^b^	0.456 ^b^	0.366 ^c^	0.423 ^b^	0.423 ^b^
		*p* value	0.019 *	0.022 *	0.072	0.035 *	0.035 *
	Physicians per 1000 inhabitants	R	−0.324 ^c^	−0.324 ^c^	−0.159 ^d^	−0.241 ^c^	−0.241 ^c^
		*p* value	0.114	0.114	0.447	0.246	0.246

Note: The Shapiro–Wilk normality test was applied, showing that the count of leishmaniasis cases in the period 2017–2021 was not normally distributed, so that Spearman’s Rho statistic was used. *: The results are significant (*p* < 0.05); ^a^ good direct correlation (+) or inverse correlation (−); ^b^ moderate direct correlation (+) or inverse correlation (−); ^c^ low direct correlation (+) or inverse correlation (−); ^d^ minimum direct correlation (+) or inverse correlation (−).

**Table 2 tropicalmed-08-00484-t002:** Descriptive statistics of factors associated with leishmaniasis cases in Peru, 2021.

Factor	Indicator	Cluster (µ ± σ)			
		C1	C2	C3	C4
		(n = 9)	(n = 3)	(n = 4)	(n = 9)
	Confirmed leishmaniasis cases 2021	73.8 ± 107	352 ± 231	207 ± 85.1	230 ± 210
Climatic and environmental	Total annual precipitation	141 ± 309	2556 ± 493	1249 ± 538	674 ± 366
Social	Households with basic access to infrastructure	86.5 ± 3.91	43.4 ± 2.46	64 ± 6.36	73.3 ± 6.98
	Households with drinking water	95.3 ± 1.47	75.4 ± 12.3	85.5 ± 4.52	89.8 ± 6.19
	Households with access to drainage–sanitation	85 ± 5.17	43.9 ± 2.23	59 ± 9.05	67.1 ± 7.78
	Households with access to electric lighting	98.1 ± 0.92	88.6 ± 6.62	88.7 ± 4.57	94.3 ± 2.66
	Illiteracy rate of the population 15 years of age and older	4.24 ± 1.89	4.53 ± 0.907	7.85 ± 3.66	8.47 ± 2.03
	Index of regional social progress	63.2 ± 2.58	52.2 ± 1.8	55.5 ± 1.8	57.9 ± 1.12
	Unsatisfied basic needs	19.6 ± 4.75	46.9 ± 10.6	41.9 ± 4.57	31.7 ± 4.11
Demographic	Population > 64 years old	10.1 ± 1.05	5.83 ± 0.153	6.1 ± 0.542	7.27 ± 0.819
	% Urban population	83.3 ± 10.9	71.7 ± 7.55	52.5 ± 13.3	54.5 ± 21.9
Health	Mortality due to transmissible diseases per 100,000 inhabitants	113 ± 15.3	247 ± 45	155 ± 11.9	130 ± 20.9

Note: C1 (9 regions): Ancash, Arequipa, Callao, Ica, La Libertad, Lambayeque, Lima, Moquegua, and Tacna; C2 (3 regions): Loreto, Madre de Dios, and Ucayali; C3 (4 regions): Amazonas, Huánuco, Pasco, and San Martín; C4 (9 regions): Apurímac, Ayacucho, Cajamarca, Cusco, Huancavelica, Junín, Piura, Puno, and Tumbes.

## Data Availability

The research data is available upon request of interested parties.

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
