# Peer review of "Cluster Analysis of Factors Associated with Leishmaniasis in Peru"

_tropicalmed, 2023, doi:10.3390/tropicalmed8110484_

Round 1

Reviewer 1 Report

1. Please add a literature review to your introduction.

Some examples. You can find and add other studies:

https://www.ncbi.nlm.nih.gov/pmc/articles/PMC3339126/

https://www.tandfonline.com/doi/abs/10.1179/2047773215Y.0000000032?journalCode=ypgh20

2. What is your study novelty?  Which research gaps did you cover in your document? Add them to the introduction.

3. Please add more maps to your document. For example, rate maps.

4. Please explain used analysis techniques in detail.

5. Please clarify the data of your dependent and independent variables. For example, how is annual average temperature data extracted? For where? Explain other factors data in detail.

6. Why do you use cluster analysis? And, why didn't you use geographical regression analysis methods or other spatial correlation methods? Why don't you use other geographical cluster analysis methods?

7. Please number each sub-figures and cite them in your text.

8. Please remove the journal’s predefined paragraph from your discussion part.

9. I think it is better to rewrite the conclusion section. See more perfect papers.

Author Response

1, 2.- The introduction has been modified, adding the suggested literature plus the novelty of the study.

3.- It is not possible to add more figures or tables in the document, because the guidelines of the journal indicate a quantity limit. 

4, 5.- In the material and methods section, the statistical analysis techniques used, and how the data have been extracted, have been explained in detail.
6.-Both cluster-analysis and geographically-weighted regression are techniques used in spatial analysis and geographical statistics. However, they are used for slightly different purposes and may be more appropriate in certain contexts, depending on the nature of the data and the objectives of analysis. Considerations to take into account are:
Pattern discovery and segmentation: Cluster-analysis is used to group similar observations into groups (clusters) based on their characteristics. It is used to discover spatial patterns and segment data into homogeneous groups, cluster-analysis is the most appropriate.

Initial data exploration: Cluster analysis is a powerful tool for exploring data and understanding the relationships between variables in a spatial context. It helps to identify areas with similar characteristics and provide valuable information for future decisions. Since our main goal is to get an overview of the spatial patterns in your data, cluster analysis is more effective.

Simplicity and communication of results: Cluster analysis tends to generate more easily interpretable and visual results in the form of maps of spatial clusters. This is useful for communicating findings to a non-technical audience or for making initial decisions based on general patterns.

7. Subfigures have been numbered and cited in the text.

8. The journal’s default paragraph has been removed in the discussion part.

9.The conclusions have been detailed in response to the objectives of the study.

Reviewer 2 Report

the introduction has to be reformulated because, for example, the data on line 33 (30 species) does not match those on line 40 (90 species of vectors)
on line 43 it says that the main reservoirs are dogs this data is considered only true for visceral leishmaniasis especially in cases of urban and periurban leishmaniasis, please check the literature
in lines 44-45 and saying that the human is the main reservoir, but this is mainly true for lesihmania donovani and for the case of post-Kalazar patients, please review the literature
on line 46 I think incubation is not the correct word
On lines 49-50 the authors could explain better, because in Europe for example visceral leishmaniasis is caused by Leishmania infantum and in the Indian subcontinent (Asia) it is caused by Leishmania donovani, please check the updated literature
in line 60 for example the term is "neglected disease" no "least attended pathologies"
in the paragraph of line 60-71 please point out that in Peru the visceral form does not exist and therefore the human does not behave as a reservoir
on lines 78-79 pleaseadd bibliography for the peercentages presented, also discuss about other species that are present in Peru
in the introduction the authors could disscus mmore facts about the vectors in Peru.
With respect to the materials and methods the actors could reformulate the same. for example if there was a filtering of the data (for example by type of diagnosis). It is not explained where the authors obtained the data temperature, climate, humidity etc.
in the statistical analysis line 148 it is not just because the variables do not have a normal distribution that Spearman correlation is used for quantitative variables vs ordinal qualitative variables, which is the case of the present study
clustering methods are also not described
in table 1 the authors could classify the correlations by type/strength for example 03 is weak correlation
The authors can also make a pipeline of the statistical methods used, including exploratory ones.

the authors must improve the text

Author Response

The entire introduction has been rephrased, as suggested by the reviewer.

In the analysis method, the techniques used have been explained in detail.

The correlations have been interpreted as suggested.

Reviewer 3 Report

See the attachment.

Moderate editing of English language is required.

Author Response

1.- The wording of the objective of the study in the introduction was improved.

2,3, 4.- The introduction has been modified, taking into account what has been suggested, with appropriate literature and the originality of the study, and taking into account other similar studies.

5, 6.- Figures 4 and 5 have been described in detail. We consider that the figures in the article are those necessary to explain the work. We cannot add others, since the maximum limit of figures and tables indicated in the journal guidelines has already been reached.

7.- In the discussion there is a comparison of differences or similarities between the results obtained with those of other studies.

8.-The resolution of figures, as required in the journal guidelines, has been taken into account.

9,10, 11.- In the discussion, the results obtained have been to related with other articles and real-life scenarios. Also, in the final part there is indication of how this study contributes to health policies in Peru.

Round 2

Reviewer 1 Report

Thank you for your revisions. 

I just have one minor revision recommendation. The maps in Figure 6 are extremely basic and simple. It is preferable to use ones with cartographic features (North arrow, scale and etc.) that are professional. 

Thank you.

Author Response

Figure 6 has been changed according to the reviewer's suggestion

Reviewer 2 Report

in line 74 is leishmania species  not leishmaniasis species

is better if the authors write de genus and the specie

for leishmania donovani and other species them human is nor the main resevoir etc

in line 101 is not least-treated is neglected disease

in line 147 again the human is not the resservoir

in line 154 is Lutzomyia not Lu in line 163 too

reformulate paragraph beyween lines 188-192

in line 197-198 the authors are sure that the study was conducted in madrid?

in line 207 is for instead from i think

must revise the english

Author Response

1.- In line 74, leishmaniasis was changed to the Leishmania species.
2.- It has been shown in other studies that humans are the main reservoir of Leishmania donovani : https://www.sciencedirect.com/science/article/pii/S0001731021001083

https://www.ncbi.nlm.nih.gov/pmc/articles/PMC9609364/

3.- On line 100 the wording was changed mentioning that the disease is neglected in the world
4.- Line 147 has been corrected, eliminating that the human is the reservoir
5.- Lines 154 and 163 were corrected, replacing Lu with Lutzomyia
6.- The paragraph between lines 188-192 was reformulated
7.- It was verified that the study was carried out in Spain and line 207 was followed as suggested.
8.- The changes are highlighted in yellow in the document.
9.- The manuscript has been reviewed by a native translator.

Round 3

Reviewer 2 Report

in line 72  please put Leishmania donovani and not just Donovani

in line 73 the are not the main reservoir please check and insert another reference

in line 77 please reformulate the phrase " the clasification of this pathology is based on the región" the pathology does not depend on the geographical area

in line 83 the word "incubation " is no t the correct word

in line 85 mucocutaneous manifestation does not necessarily occur within 2 years , please change this phrase in line 85

in line 86 is not clear wich type of clinical form of leishmaniasis the authors are talking about

in line 92 actually is consensus that is leishmania infantum chagasi and not only leishmana chagasi actually leishmania chagasi and leishmania infantum are considered the same species

in line 93 please chage the phrase " is dogs"

in lines 93-94 isnot clear what type of leishmaniasis the authors are talking about

in line 100 wich disease the authors are talking about? is not clear

in line 110 the authors can talk about global climatic change instead climate change only

IN line 106 please check the phrase Higlighting that

in line 145 please change the phrase " are known"

in line 161 will be better if the authors use the term " clinical manisfestation" instead type of leihmaniasis

in line 169 not only insects are considered vectors of leishmaniasis

in line 190 the word contracting is not the correct word

in line 201 be more specific than the phrase "latin america"

in line 204 What is the region the authors are referring to?

in line 222 please insert Callao región

The authors can use another word instead "departament" to describe an geographical administrated area, or explain what the department word means in terms of administration or geographical area

in the statistical analysis line it is not just because the variables do not have a normal distribution that Spearman correlation is used for quantitative variables vs ordinal qualitative variables, which is the case of the present study
clustering methods are also not described
in table 1 the authors could classify the correlations by type/strength for example 03 is weak correlation
The authors can also make a pipeline of the statistical methods used,

including exploratory ones.the exploratory methods cand explain the rationale of the study

check the english please  specially some words that used in a wrong context

Author Response

Dear reviewer, we appreciate your comments that aim to improve the manuscript. We have addressed each of them and improved according to what was suggested by you.

On line 72, now line 73, Leishmania donovani” was placed instead of just “Donovani
On line 73, now line 74, was corrected not stating that they are the main deposit
Lines 77 to 81 were eliminated and it was clarified in line 84 that this pathology varies according to the region in which it occurs.
On line 83, now line 79, the word "incubation" was replaced with latency period
Line 85: Mucocutaneous manifestation does not necessarily occur within 2 years; this was removed.
Lines 86,  92, 93 and 94 were modified.
On line 100, now line 95, we talk about leihsmaniasis
On line 110, now line 105, the authors modify and put global climate change instead of just climate change
On line 106, now line 101, we change the phrase Highlighting that
on line 145, now line 140, we change the phrase "are known"
On line 161, now line 156, the authors use the term "clinical manifestation" instead of type of leishmaniasis.
In line 169, now line 164, it was modified that not only insects are considered vectors of leishmaniasis
On line 190, now line 185, the word hiring was changed
In lines 201 and 204, now lines 196-199, the study areas were specified
In line 222, now line 216, Callao region was inserted
The word region was used instead of "department" to describe an administered geographic area. In the statistical part, the grouping methods and statistical methods used in the analysis were described. In Table 1, the authors classified the correlations by type/strength.